# A Nucleus Accumbens Tac1 Neural Circuit Regulates Avoidance Responses to Aversive Stimuli

**DOI:** 10.3390/ijms24054346

**Published:** 2023-02-22

**Authors:** Zi-Xuan He, Ke Xi, Kai-Jie Liu, Mei-Hui Yue, Yao Wang, Yue-Yue Yin, Lin Liu, Xiao-Xiao He, Hua-Li Yu, Zhen-Kai Xing, Xiao-Juan Zhu

**Affiliations:** Key Laboratory of Molecular Epigenetics, Institute of Genetics and Cytology, Northeast Normal University, Ministry of Education, Changchun 130021, China

**Keywords:** nucleus accumbens, tachykinin precursor 1, aversion

## Abstract

Neural circuits that control aversion are essential for motivational regulation and survival in animals. The nucleus accumbens (NAc) plays an important role in predicting aversive events and translating motivations into actions. However, the NAc circuits that mediate aversive behaviors remain elusive. Here, we report that tachykinin precursor 1 (Tac1) neurons in the NAc medial shell regulate avoidance responses to aversive stimuli. We show that NAc^Tac1^ neurons project to the lateral hypothalamic area (LH) and that the NAc^Tac1^→LH pathway contributes to avoidance responses. Moreover, the medial prefrontal cortex (mPFC) sends excitatory inputs to the NAc, and this circuit is involved in the regulation of avoidance responses to aversive stimuli. Overall, our study reveals a discrete NAc Tac1 circuit that senses aversive stimuli and drives avoidance behaviors.

## 1. Introduction

Reward and aversion are critical for motivated behaviors and are associated with many mood disorders. Unexpected stimuli and threats drive aversive behaviors, an innate response crucial to the survival of animals [1]. Aversive stimuli engage negative emotions and contribute to prominent psychiatric disorders. Enormous advances have been made in understanding the neural circuits underlying reward [2,3,4,5,6,7]. However, the neural circuits underlying aversion remain elusive.

It is widely thought that the nucleus accumbens (NAc) is a critical brain region in the reward and aversion circuits that integrate different inputs, leading to motivated behaviors [8,9,10,11,12,13,14]. Anatomically, the NAc can be divided into the core, lateral shell, and medial shell [15]. It has been found that distinct NAc neural circuits are involved in different brain functions [16,17,18,19,20,21]. Dopamine transmissions from the ventral tegmental area have been linked to reward and aversion processing [18]. Glutamatergic inputs from the thalamic paraventricular nucleus to the NAc regulate aversion [19,22]. How the NAc regulates opposite behaviors at the same time remains elusive. Thus, it is worth investigating whether distinct NAc subregions are included in discrete neural circuits involved in aversion.

The major projection neurons in the NAc are medium spiny neurons (MSNs), distinguished by their dopamine receptor expression (D1-MSNs and D2-MSNs) [23,24,25]. Markers for D1-MSNs and D2-MSNs also include the expression of different peptides [26,27]. Substance P, the major peptide encoded by tachykinin precursor 1 gene (TAC1), and dynorphin are exclusively expressed in D1-MSNs [28,29,30]. Previous work demonstrated that dynorphin-containing neurons in the NAc mediate negative affective states [16,31,32]. This raises the possibility that Tac1 neurons in the NAc medial shell may be involved in the regulation of aversion.

The NAc has received attention as a crucial convergence point of reward and aversion circuits, as it receives multiple projections from the ventral tegmental area (VTA), medial prefrontal cortex (mPFC), basolateral amygdala (BLA), and hippocampus [33,34]. The mPFC is strongly related to neural circuits encoding aversion and decision making [35,36,37]. The prelimbic and infralimbic regions of the mPFC have been implicated in aversion [38,39,40]. However, studies have yielded conflicting findings. How the mPFC regulates aversion through specific neural circuits remains underexplored.

Here, we show that tachykinin precursor 1 (Tac1) neurons in the NAc medial shell mediate avoidance responses to aversive stimuli. Neural tracing and electrophysiological data show that NAc^Tac1^ neurons project inhibitory signals to the lateral hypothalamic area (LH) and modulate avoidance behavior in the presence of aversive stimuli. Additionally, neurons in the NAc medial shell receive inputs from mPFC glutamatergic (mPFC^Glut^) neurons, and optogenetic manipulation of the mPFC^Glut^ → NAc circuit regulates aversive behaviors. These results indicate the essential role of Tac1 neurons in encoding aversive stimuli and regulating behavioral responses.

## 2. Results

### 2.1. NAc^Tac1^ Neurons Regulate Avoidance Behavior in Response to Aversive Stimuli

To investigate the expression of Tac1 neurons in the NAc, we crossed the *Tac1*-internal ribosome entry site 2 (IRES2)-Cre mouse line [41] with a Cre-dependent tdTomato reporter line, Ai9 [42] (Figure 1A). We observed that Tac1-tdTomato cellular expression closely matched endogenous substance P and Dopamine Receptor 1 (Figure 1B–G). To mimic aversion in mice, *Tac1*-Cre male mice were given an injection of formalin in the plantar surface of a hindpaw, as previously described [43,44]. Patch-clamp recordings were performed on Tac1 neurons in the NAc (Figure 1H). We observed decreased excitability of Tac1 neurons in the medial shell, but not in the lateral shell (Figure 1I,J and Appendix A). These data indicate that Tac1 neurons in the NAc medial shell are involved in the circuit regulating aversion.

To determine whether NAc^Tac1^ neurons in the NAc medial shell regulate aversive behaviors, we performed chemogenetics using designer receptors exclusively activated by designer drugs (DREADDS). To selectively manipulate the activity of Tac1 neurons, we bilaterally injected AAV-DIO-hM3D(Gq)-mCherry, AAV-DIO-hM4D(Gi)-mCherry, and AAV-DIO-mCherry into the NAc medial shell of *Tac1*-Cre male mice (Figure 1K). Formaldehyde has been shown to act as an unfamiliar aversive stimulus for rodents without altering their motor activity [45]. We thus measured the approach-avoidance behaviors of male mice to an aversive stimulus (formaldehyde) while inhibiting or activating the activity of NAc^Tac1^ neurons. A piece of cotton dipped in 5% formaldehyde was placed on one side of a three-chamber arena. Mice tend to explore a novel object, but animals display strong avoidance behaviors when exposed to formaldehyde. Mice were introduced into the chamber containing formaldehyde. Interactions with formaldehyde were recorded for 5 min. The opposite chambers of the arena were designated the ‘safe’ area and ‘center’ area. We observed that hM3D(Gq)-injected mice spent significantly more time exploring the aversive stimulus than hM4D(Gi)- and mCherry (control)-injected mice (Figure 1L–O). To investigate whether the activity of Tac1 neurons regulates interactions with a neutral stimulus in mice, we performed the approach experiment and replaced the piece of 5% formaldehyde cotton with a piece of regular cotton. We found that the activity of NAc Tac1 neurons did not affect time spent interacting with neural stimuli (Appendix A). Moreover, olfaction and locomotion were not affected by hM4D(Gi) or hM3D(Gq) injection (Appendix A). These results suggest that NAc^Tac1^ neurons in the medial shell are crucial to avoidance behaviors in response to aversive stimuli.

### 2.2. NAc^Tac1^ Neurons Project to the LH

To identify possible downstream targets of NAc^Tac1^ neurons that may encode aversive stimuli, we injected AAV-DIO-mCherry into the NAc medial shell of *Tac1*-Cre mice. Four weeks later, the animals were euthanized, and the distribution of neurons that NAc^Tac1^ neurons target in the brain was examined (Figure 2A–C). The whole-brain mapping results indicated that dense mCherry-labeled terminals were found in the lateral hypothalamic area (LH) (Figure 2D–F, Appendix A).

We next assessed the synaptic function of NAc^Tac1^ neurons projecting to the LH. We first expressed channel rhodopsin-2 (ChR2) in NAc^Tac1^ neurons, and then, selectively activated the terminals of NAc^Tac1^ neurons in the LH via optogenetic stimulation (5 ms pulse, 20 Hz) (Figure 2G,H). In the whole-cell patch-clamp configuration, inhibitory postsynaptic currents (IPSCs) were recorded in 17 out of 42 LH neurons (Figure 2I). However, no excitatory postsynaptic currents were recorded. IPSCs were eliminated via pretreatment with the GABA-A receptor antagonist bicuculline (Figure 2J,K, Appendix A). These results suggest that NAc^Tac1^ neurons send inhibitory inputs to the LH.

### 2.3. NAc^Tac1^-to-LH Projection Mediates Avoidance Behaviors in Response to Aversive Stimuli

To assess whether the NAc^Tac1^→LH circuit regulates avoidance responses to aversive stimuli. Male *Tac1*-Cre mice were unilaterally injected with AAV-DIO-mCherry and AAV-DIO-ChR2-mCherry (Figure 3A,B). Six weeks later, we carried out an approach-avoidance assay for evaluation (Figure 3C). A piece of cotton dipped in 5% formaldehyde was placed in one corner of a square chamber. Mice were introduced into the chamber. Their interactions with formaldehyde were recorded. Compared with the control stimulation, the selective delivery of blue light (5 ms pulse, 20 Hz for 5 min) to the LH of ChR2-expressing terminals elicited a significant increase in interaction time with formaldehyde (Figure 3D,E). We also calculated the total distance traveled by the mice in the arena and found that locomotion was not affected (Figure 3F). 

We then selectively inhibited the NAc^Tac1^ terminals in the LH by delivering continuous yellow light to the LH of male mice bilaterally infected with AAV-DIO-NpHR-eYFP in the NAc medial shell (Figure 3G,H). In the approach-avoidance assay, the photoinhibition of NAc^Tac1^→LH projection significantly decreased interaction time with formaldehyde without affecting locomotion (Figure 3I–L). We also carried out a real-time place aversion (RTPA) assay and found that the photoinhibition of NAc^Tac1^→LH projection elicited avoidance of the photoinhibition-paired chamber (Appendix A). Taken together, these data indicate that the NAc^Tac1^→LH circuit is crucial to avoidance behaviors in response to aversive stimuli.

### 2.4. mPFC^Glut^ Inputs Activate NAc Neurons

Next, we sought to identify upstream brain regions of NAc^Tac1^ neurons that might mediate aversive behaviors. We employed a monosynaptic viral tracing strategy in *Tac1*-*Cre* mice. The NAc medial shell of *Tac1*-Cre mice was injected with AAV-DIO-RVG and AAV-DIO-TVA-GFP. Four weeks later, RV-EnVA-dsRed was injected into the LH (Figure 4A–**C**). We found that the medial prefrontal cortex (mPFC) was projected to NAc^Tac1^ neurons (Figure 4D). Based on emerging studies [37,38,46,47,48] showing that the mPFC is critical for neural circuits of aversion, we focused on neurons in the mPFC projecting to NAc^Tac1^ neurons. To determine the kinds of mPFC neuron that are involved in the NAc^Tac1^ circuit, we performed immunofluorescence experiments and found that mCherry-labeled neurons co-expressed the glutamatergic marker VGLUT2 (Figure 4E and Appendix A).

We next evaluated the synaptic function of mPFC^Glut^ neurons projecting to NAc neurons. We first expressed ChR2 in mPFC^Glut^ neurons by injecting AAV-CaMKIIα-ChR2-eYFP into the mPFC, and then, selectively activated NAc neurons that were receiving projections from mPFC^Glut^ neurons via optogenetic stimulation (5 ms pulse, 20 Hz) (Figure 4F,G). In the whole-cell patch-clamp configuration, excitatory postsynaptic currents (EPSCs) were recorded in 32 out of 60 NAc neurons (Figure 4H). However, no inhibitory postsynaptic currents were recorded. EPSCs were eliminated via pretreatment with the AMPA receptor antagonist CNQX (Figure 4I,J and Appendix A). To identify whether Tac1 neurons in the NAc medial shell received inputs from the mPFC^Glut^, we expressed ChR2 in mPFC^Glut^ neurons in *Tac1*-Cre; Ai9 mice. In the patch-clamp recording, EPSCs were recorded in Tac1 neurons in the NAc medial shell (Appendix A). These results suggest that mPFC^Glut^ neurons project excitatory signals to Tac1 neurons in the NAc medial shell.

### 2.5. The mPFC^Glut^-to-NAc Circuit Modulates Avoidance Behaviors in Response to Aversive Stimuli

To investigate whether activation of the mPFC^Glut^ →NAc circuit decreases avoidance behaviors in response to aversive stimuli. Male mice were unilaterally injected with AAV-CaMKIIα-eYFP and AAV-CaMKIIα-ChR2-eYFP (Figure 5A,B). Six weeks later, we carried out an approach-avoidance assay for evaluation (Figure 5C). Compared with the control stimulation, the selective delivery of blue light (5 ms pulse, 20 Hz for 5 min) to ChR2-expressing terminals in the NAc medial shell elicited a significant increase in interaction time with formaldehyde (Figure 5D,E). We also calculated the total distance traveled by mice in the arena and found that locomotion was not affected (Figure 5K).

Next, we selectively inhibited mPFC^Glut^ terminals in the NAc medial shell by delivering continuous yellow light to the LH of male mice bilaterally infected with AAV-CaMKIIα-eNpHR3-mCherry in the mPFC (Figure 5G,H). In the approach-avoidance assay, inhibition of the mPFC^Glut^ →NAc pathway significantly reduced interaction time with formaldehyde without affecting locomotion (Figure 5I–L). We also carried out a real-time place aversion (RTPA) assay and found that inhibition of the mPFC^Glut^ →NAc pathway elicited avoidance of the photoinhibition-paired chamber (Appendix A).

Furthermore, we determined whether the activation of mPFC^GLUT^ neurons could attenuate aversive behaviors following the inhibition of NAc^Tac1^ neurons. AAV-CaMKIIα-hM3D(Gq)-mCherry or AAV-CaMKIIα-mCherry was injected in the mPFC, while AAV-DIO-hM4D(Gi)-mCherry was injected in the NAc (Appendix A). We found that the activation of the mPFC neurons was able to attenuate avoidance behaviors in response to aversive stimuli (Appendix A). Taken together, these data indicate that the mPFC^Glut^ →NAc circuit is crucial to avoidance behaviors in response to aversive stimuli.

## 3. Discussion

Using neural circuit tracing, chemogenetics, electrophysiology, and optogenetics approaches, we found that NAc^Tac1^ neurons in the medial shell mediate avoidance responses to aversive stimuli in 10–14-week-old male mice. NAc^Tac1^ neurons send inhibitory inputs to the LH, and the Nac^Tac1^→LH circuit is required for aversive behaviors in mice. Moreover, Nac neurons receive glutamatergic inputs from mPFC^Glut^ neurons, and mPFC^Glut^→Nac projection regulates behavioral responses in the presence of aversive stimuli.

Our mapping study of output circuits demonstrated that Nac^Tac1^ neurons in the medial shell project to LH neurons. The LH is a brain region that contains heterogeneous cell populations [49] and is involved in the regulation of multiple behaviors, such as feeding, aversion, and reward-seeking [50,51]. Previous studies have reported that the activation of LH neurons causes avoidance and aversive behaviors [52,53,54]. This is consistent with our data, which show that the optogenetic inhibition of NAc^Tac1^ terminals in the LH induced aversive behaviors in mice. However, the LH receives multiple excitatory and inhibitory inputs from both cortical and subcortical structures, further research is needed to fully resolve the neuron populations receiving inhibitory inputs from NAc^Tac1^ neurons and the mechanisms underlying aversion in the LH.

Previous studies have reported that mPFC neurons project to the NAc and that these neurons are able to elicit avoidance [55]. It has also been reported that projections from the mPFC to the NAc have no effect on aversion [37]. These conflicting results may partially be caused by the heterogeneity of the different regions of the NAc: the core, medial shell, and lateral shell. These regions contain similar classes of medial spiny projection neurons (MSNs). NAc medial shell MSNs have been described as “medium-small spiny neurons” with low density [56]. In addition, the NAc medial shell shows a perplexing phenotype that opposes the classical direct and indirect pathway model [16,57,58]. In addition, previous work indicates that D1 neurons in the NAc also represent a portion of the classical indirect pathway and are activated by aversive stimuli [59,60]. Taken together, the classical striatal direct and indirect pathway models are not applied to the NAc. In this study, we found that mPFC^Glut^ neurons send excitatory signals to neurons in the NAc medial shell. Moreover, this mPFC^Glut^→NAc circuit is involved in the regulation of aversion. However, aversion is a multidimensional construct, and we cannot rule out the possibility that other neurons in the NAc receive inputs from the mPFC and contribute to aversive behaviors. 

In summary, we delineated distinct Tac1 neurons as encoding aversive stimuli. Furthermore, we dissected the dedicated function of this circuit and identified it as a critical component of the aversion circuit. These results may improve our understanding of the aversion circuit. By understanding the structure and mechanisms underlying aversion and negative prediction, it will be possible to design intervention strategies for pathological depressive conditions.

## 4. Materials and Methods

### 4.1. Animals

All experimental procedures were approved by the Animal Advisory Committee of Northeast Normal University, China. The laboratory was kept under specific pathogen-free (SPF) conditions. All mice were maintained on a 12–12 h light–dark cycle (lights on from 6:00 to 18:00 every day), with food and water provided ad libitum. All behavioral tests were performed during the light period. C57BL/6J mice were obtained from Huafukang Animal Center, Beijing, China. *Tac1*-IRES2-Cre mice (Jax No. 021877) were obtained from Jackson Laboratory (USA). Ai9 mice (Jax No. 007905) were kindly provided by Prof. Chunjie Zhao from Southeast University.

### 4.2. Viral Vector Generation

For monosynaptic tracing, AAV-EF1α-DIO-His-EGFP-2a-TVA (AAV2/9, 5.53 × 10^12^ particles mL^−1^), AAV-EF1α-DIO-RG (AAV2/9, 5.22 × 10^12^ particles mL^−1^), and RV-ENVA-ΔG-dsRed (3.10 × 10^8^ particles mL^−1^) were purchased from BrainVTA (Wuhan, China). AAV-EF1α-DIO-mCherry (AAV2/9, 1.47 × 10^13^ particles mL^−1^) was purchased from GeneChem (Shanghai, China).

For functional analysis, AAV-EF1α-DIO-hM4D(Gi)-mCherry (AAV2/9, 1.044 × 10^12^ particles mL^−1^), AAV-EF1α-DIO-hM3D(Gq)-mCherry (AAV2/9, 2.205 × 10^12^ particles mL^−1^), and AAV-EF1α-DIO-ChR2-mCherry (AAV2/9, 1.25 × 10^13^ particles mL^−1^) were purchased from GeneChem (Shanghai, China). AAV-CaMKIIα-ChR2(H134R)-eYFP-WPRE-hGH polyA (AAV2/9, 2.77 × 10^12^ particles mL^−1^), AAV-CaMKIIα-eYFP-WPRE-hGH polyA (AAV2/9, 6.6 × 10^12^ particles mL^−1^), AAV-CaMKIIα-eNpHR3.0-mCherry-WPRE-hGH polyA (AAV2/9, 4.25 × 10^12^ particles mL^−1^), and AAV-CaMKIIα-mCherry-WPRE-hGH polyA (AAV2/9, 2.29 × 10^12^ particles mL^−1^) were purchased from BrainVTA (China).

### 4.3. Viral Tracing

For output mapping, the NAc medial shell of *Tac1*-Cre mice was injected with with AAV-EF1α-DIO-mCherry (200 nL). For input mapping, AAV-EF1α-DIO-His-EGFP-2a-TVA and AAV-EF1α-DIO-RG (1:1, total 150 nL) were injected into the NAc medial shell of *Tac1*-Cre mice. Four weeks later, 300 nL of RV-ENVA-ΔG-dsRed was injected into the LH. Thus, we only infected NAc^Tac1^ neurons in the medial shell that projected to the LH, and traced their inputs. The mice were sacrificed one week after RV injection.

### 4.4. Stereotaxic Injection

Mice were anesthetized with 1.0% sodium pentobarbital (0.1 g/kg body weight, i.p.). Viruses were delivered at a rate of 100 nL/min using a stereotaxic instrument (RWD Co, Shenzhen China) and a 5 µL syringe (Hamilton, Sigma, USA). After each injection, the syringe was left in place for 15 min, and then, slowly withdrawn. Experiments were performed at least 4–6 weeks after virus injection.

Stereotaxic coordinates were derived from the Paxinos and Franklin Mouse Brain Atlas and empirically adjusted. The coordinates for injection into the NAc medial shell (total volume of 400 nL) were +1.9 mm AP, ±0.6 mm ML, and −4.4 mm DV. The coordinates for injection into the LH (total volume of 150 nL) were −1.5 mm AP, ±0.9 mm ML, and −5.1 mm DV. The coordinates for injection into the mPFC (total volume of 400 nL) were +2.2 mm AP, ±0.3 mm ML, and −1.35 mm DV. For monosynaptic circuit tracing and the ChR2 experiment, viruses were delivered unilaterally. For other functional analysis, viruses were delivered bilaterally.

### 4.5. Implantation of Optical Fibers

Optogenetic behavioral experiments were performed as previously described [16,18,19], and optic fibers (NA: 0.37; INPER, Wuhan, China) were unilaterally (ChR2) or bilaterally (NpHR3.0) implanted over the LH (AP: −1.5 mm; ML: ±0.9 mm; DV: −4.9 mm) and NAc medial shell (AP: +1.9 mm; ML: ±0.5 mm; DV: −4.2 mm). The mice were subjected to behavioral tests after 2 weeks of recovery. For optogenetic activation experiments, both control and ChR2-injected mice were stimulated using a 20 Hz 465 nm blue laser (INPER, China) with 2–5 mW light power at the fiber tips. For optogenetic inhibition experiments, both control and NpHR-injected mice were continuously stimulated using a 589 nm yellow laser (INPER, China) with 2–5 mW light power at the fiber tips.

### 4.6. Immunohistochemistry

As previously described [61], mice were deeply anesthetized with sodium pentobarbital (0.5 g/kg, i.p.) and perfused transcardially with 0.1 M PBS followed by 4% paraformaldehyde (PFA) in PBS. Their brains were then post-fixed overnight at 4 °C and transferred to 30% sucrose solution. Sagittal and coronal sections were cut on a freezing microtome (Leica, CM 1950, USA) at a thickness of 40 µm. The sections were rinsed in PBS, and then, incubated in blocking solution (0.2% Triton X-100, 10% serum, and 2% BSA in 0.1 M PBS) for 2 h. After washing with PBS, the sections were counterstained with DAPI (1:2000, Life Technologies, D3571, USA) for 8 min. The sections were then covered with ProLong gold mounting media (Thermo Fisher, P36930, USA). The following primary antibodies were used: NeuN (1:1000; EMD Millipore, MAB377, USA), substance P (1:1000; Abcam, ab10353, USA), VGLUT2 (1:500; Synaptic Systems, 135 402, Germany), and DRD1 (1:500; Novus Biologicals, NB110-60017, USA). The following secondary antibodies were used: Alexa Fluor 488-conjugated goat anti-mouse (1:1000; Invitrogen, A21121, USA), Alexa Fluor 488-conjugated goat anti-rabbit (1:1000; Invitrogen, A11008, USA), and Alexa Fluor 488-conjugated goat anti-guinea pig (1:1000; Invitrogen, A11073, USA). All images were acquired using a Zeiss LSM 880 confocal microscope (USA).

### 4.7. Ex Vivo Electrophysiology

Mice were deeply anesthetized with sodium pentobarbital and quickly decapitated to remove their brains. Acute slices (300 μm thick) were cut using a vibrating microtome (Leica, VT 1000S). The sections were quickly transferred to a recovery chamber and incubated at 35 °C for 30 min in recovery solution comprising 93 mM NMDG, 1.2 mM NaH_2_PO_4_, 30 mM NaHCO_3_, 20 mM HEPES, 25 mM D-Glucose, 5 mM Na-ascorbate, 2 mM Thiourea, 3 mM Na-pyruvate, 3 mM KCl, 10 mM MgSO_4_, 0.5 mM CaCl_2_, 93 mM HCl, and 12 mM NAC (pH 7.4). The slices were then incubated at room temperature for 1 h in carbogenated artificial cerebral spinal fluid (aCSF) comprising 120 mM NaCl, 2.5 mM KCl, 1.0 mM NaH_2_PO_4_, 26 mM NaHCO_3_, 11 mM D-glucose, 2.0 mM MgCl_2_, and 2.0 mM CaCl_2_ (pH 7.4) before recording. Recordings were made at 33 °C (TC-324B; Warner Instruments, USA). All solutions were saturated with 95% O_2_/5% CO_2_.

Whole-cell patch-clamp recordings were performed using an EPC-10/2 amplifier (HEKA, Germany). The recording pipettes were pulled from borosilicate glass tubes (Sutter Instruments, USA) and had a resistance of 3–6 MΩ; only whole-cell patches with a series resistance < 15 MΩ were used for recordings. EPSC and IPSC were recorded by holding the membrane potential at −70 mV.

For optical recording in the LH, AAV-DIO-ChR2-mCherry was injected into the NAc medial shell of *Tac1*-Cre mice, and LH neurons in areas with a high density of mCherry terminals were patched. ChR2 with 465 nm blue light was delivered via a laser (INPER-B1–465, INPER, China). To record optically evoked IPSCs (oIPSCs) in LH neurons, CNQX (50 µM, Tocris Bioscience, 1045, USA) was added to the aCSF. Patch pipettes were filled with 135 mM CsCl, 1 mM EGTA, 4 mM Mg-ATP, 0.6 mM Na-GTP, and 10 mM HEPES (pH 7.4).

For optical recording in the NAc medial shell, AAV-CaMKIIα-ChR2-mCherry was injected into the mPFC of C57 mice, and NAc medial shell neurons in the areas with a high density of mCherry terminals were patched. ChR2 with 465 nm blue light was delivered via a laser (INPER-B1–465, INPER, China). To record optically evoked EPSCs (oEPSCs) in NAc medial shell neurons, bicuculline (20 µM, Tocris Bioscience, 0130) was added to the aCSF. Patch pipettes were filled with 130 mM K-gluconate, 1 mM EGTA, 5 mM Na-phosphocreatine, 2 mM Mg-ATP, 0.3 mM Na-GTP, and 10 mM HEPES (pH 7.4).

Data were acquired using PATCHMASTER 1.3 (HEKA, Germany) and analyzed using MiniAnalysis 1.0 (Synaptosoft), Clampfit 10.0 (Molecular Devices), and Igor 5.03 (Wavemetrics) software.

### 4.8. Behavioral Assays

All mice used for the behavioral assays were male mice and their littermates. An experimenter blinded to the genotypes performed all the tests.

### 4.9. Approach-Avoidance Test

The avoidance test was conducted to measure avoidance of an unfamiliar aversive stimulus. For the chemogenetics experiments, control-, hM3D(Gq)-, and hM4D(Gi)-injected mice were i.p. injected with clozapine N-oxide (CNO; 5 mg/kg or JHU37160; 0.5 mg/kg), and introduced into the chamber half an hour later. The chamber (70 cm × 70 cm) contained three sides (safe, center, and form). A piece of cotton dipped in 5% formaldehyde was placed on the form side. The opposite sides of the chamber were designated the ‘safe’ area and ‘center’ area. As previously described [18], mice tend to explore a novel object, but animals display strong avoidance behaviors when exposed to formaldehyde. Interactions with formaldehyde were recorded and analyzed by the EthoVision XT system (Noldus, Wageningen, The Netherlands).

For the optogenetics (ChR2 and NpHR) experiments, after recovery from the surgery for virus injection and fiber implantation, mice were introduced into an arena (40 cm × 40 cm). A piece of cotton dipped in 5% formaldehyde was place in a corner of the arena. Interactions with formaldehyde were recorded in 5 min segments and analyzed using the EthoVision XT system (Noldus, Wageningen, The Netherlands). 

### 4.10. RTPA Assay

On the day of habituation, the mice were introduced into a Plexiglas box with two chambers (30 cm × 30 cm × 50 cm each) and allowed to explore the chamber freely for 15 min. One chamber was randomly designated the stimulation side, and the other was designated the non-stimulation side. The time spent in each of the chambers was recorded. Mice that spent more than 60% of the total time in either compartment were excluded from the experiments. On the day of the experiment, the mice were randomly introduced into either chamber and received continuous 589 nm yellow light (or 20 Hz 465 nm blue light) every time they entered the stimulation chamber until they moved into the non-stimulation chamber. The time spent in each chamber was recorded and analyzed using the EthoVision XT system.

### 4.11. Olfaction Test

Before the test, all pellets were removed from the home cage, but the water bottle was kept in place. On the day of the experiments, a mouse was introduced into a clean cage containing clean bedding with a depth of 3 cm. The animal was allowed to explore the arena freely for 5 min. The animal was then transferred to an empty clean cage. In the cage containing the bedding, food was buried approximately 1 cm beneath the surface in a random corner. The surface of the bedding was smoothed out, and the animal was reintroduced into the cage. The latency to find the buried food was recorded. The food was considered uncovered when the mouse started to eat it.

### 4.12. Quantification and Statistical Analyses

All experimental procedures and data analyses were conducted in a blinded manner. The number of replicates (N or n) indicated in the figure legends refers to the number of experimental subjects independently treated in each experiment. All statistical analyses were performed in Graph Pad Prism (GraphPad Software) unless otherwise stated. For normally distributed data, a Student’s test and one-way ANOVA, followed by Scheffe’s post hoc test, were used to analyze the significance between groups. For non-normally distributed data, Mann–Whitney U-tests were used to calculate the significance between groups. All statistical data can be found in the figure legends. Statistical significance was set at * *p* < 0.05, ** *p* < 0.01, and *** *p* < 0.001. The data are presented as the means ± s.e.m.

## Figures and Tables

**Figure 1 ijms-24-04346-f001:**
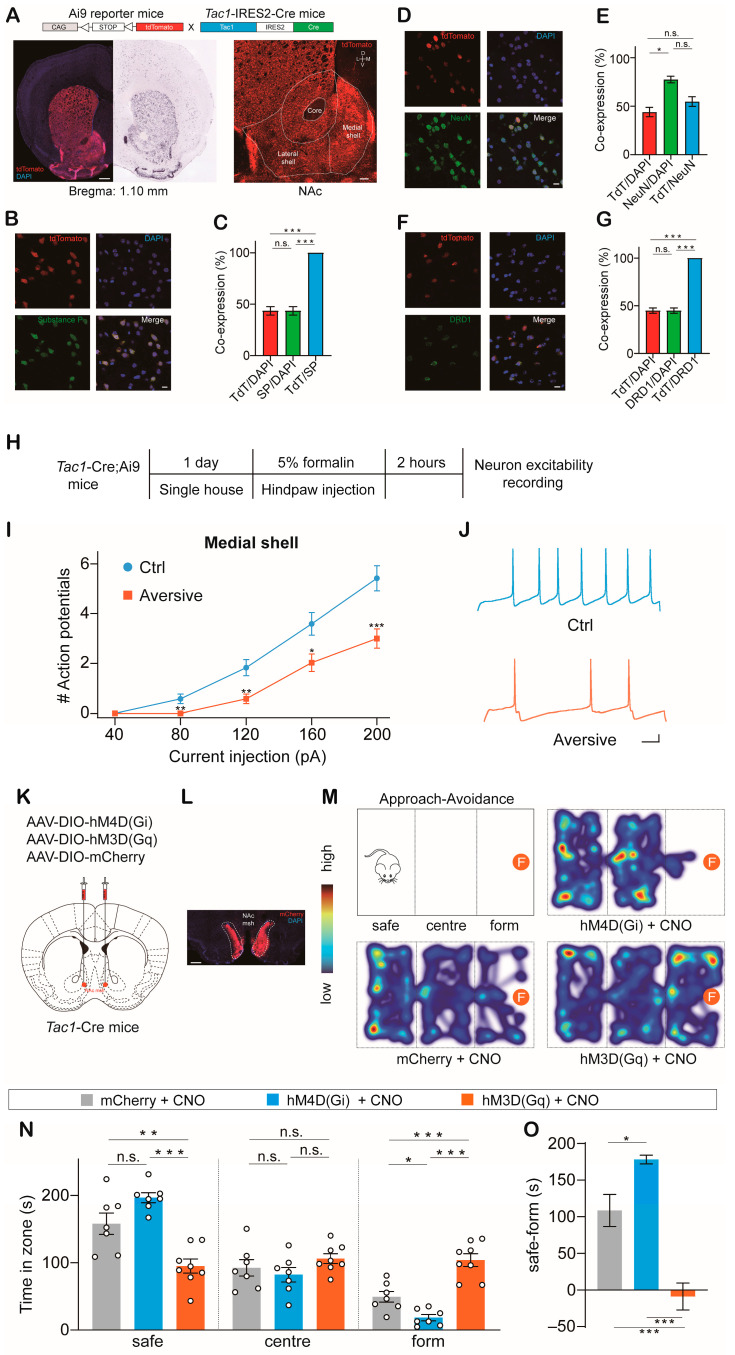
NAc^Tac1^ neurons mediate aversive behaviors: (**A**) Coronal images demonstrating the application of Tac1-IRES2-Cre and Ai9-tdTomato reporter mice to label Tac1-positive neurons in the NAc, compared to the Allen Institute for Brain Science. Scale bar: 500 µm and 100 µm, respectively. D: dorsal; V: ventral; L: lateral; M: medial. (**B**) Representative image of acute striatal slice from Tac1-Cre; Ai9 mice stained with antibodies against substance P. Scale bar: 20 µm. (**C**) Percentage of Tac1-positive neurons that co-localized with substance P; co-localization ratio (TdT/DAPI: 43.55 ± 4.096%, SP/DAPI: 43.55 ± 4.096%, TdT/SP: 100%; one-way ANOVA test, F (2,12) = 1.669, *p* < 0.0001). N = 5. (**D**) Representative image of acute striatal slice from Tac1-Cre; Ai9 mice stained with antibodies against NeuN. Scale bar: 20 µm. (**E**) Percentage of Tac1-positive neurons that co-localized with NeuN; co-localization ratio (TdT/DAPI: 43.90 ± 4.801%, NeuN/DAPI: 77.43 ± 3.455%, TdT/NeuN: 54.72 ± 5.014%; one-way ANOVA test, F (2, 9) = 14.61, *p* = 0.0015). N = 4. (**F**) Representative image of acute striatal slice from Tac1-Cre; Ai9 mice stained with antibodies against DRD1. Scale bar: 20 µm. (**G**) Percentage of Tac1-positive neurons that co-localized with DRD1; co-localization ratio (TdT/DAPI: 44.90 ± 2.846%, DRD1/DAPI: 44.90 ± 2.846%, TdT/DRD1: 100%; one-way ANOVA test, F (2,6) = 2.541, *p* < 0.0001). N = 3. Mann–Whitney U-tests were used for all panels. (**H**) Experimental timeline. (**I**) Spikes elicited in Tac1 neurons in NAc medial shell in response to current injection. (**J**) Action potentials elicited after 200 pA current injection for 500 ms (ctrl, hindpaw saline injection: N = 4, N = 12; aversive, hindpaw formalin injection: N = 4, N = 12). Scale bar: 50 ms, 10 mV. (**K**) Schematic of strategies used to express AAV-DIO-mCherry, AAV-DIO-hM3D(Gq)-mCherry, and AAV-DIO-hM4D(Gi)-mCherry in NAc medial shell of Tac1-Cre mice. (**L**) Coronal view documenting viral expression in NAc medial shell. Scale bar: 200 µm. (**M**) Schematic of avoidance assay (F: form, 5% formaldehyde solution). Control, hM3D(Gq), and hM4D(Gi) mice were i.p. injected with CNO (5 mg/kg). Heat maps display time spent in different regions of the chamber (warmer colors indicate more time). (**N**) Averaged time of control, hM4D(Gi), and hM3D(Gq) mice spent in different areas of the chamber (safe: mCherry: 158.1 ± 15.84 s, hM4D(Gi): 196.7 ± 7.395 s, hM3D(Gq): 95.21 ± 10.46 s; center: mCherry: 92.51 ± 12.12 s, hM4D(Gi): 82.33 ± 10.79 s, hM3D(Gq): 106.2 ± 7.140 s; form: mCherry: 49.42 ± 7.939 s, hM4D(Gi): 18.54 ± 4.591 s, hM3D(Gq): 104.1 ± 9.48 s. *One*-*Way ANOVA* test, safe: *F*_(2,19)_ = 19.83, *p* < 0.0001; center: *F*_(2,19)_ = 1.479, *p* = 0.2529; form: *F*_(2,19)_ = 31.43, *p* < 0.0001). (**O**) Averaged difference (time in form area – time in safe area) for control, hM4D(Gi), and hM3D(Gq) mice (one-way ANOVA test, safe – form: *F*_(2,19)_ = 31.51, *p* < 0.0001). mCherry + CNO: N = 7; hM4D + CNO: N = 7; hM3D + CNO: N = 8. N: animal number. NAc msh: nucleus accumbens medial shell. All data are means ± s.e.m. * *p* < 0.05; ** *p* < 0.01; *** *p* < 0.001; n.s.: not significant.

**Figure 2 ijms-24-04346-f002:**
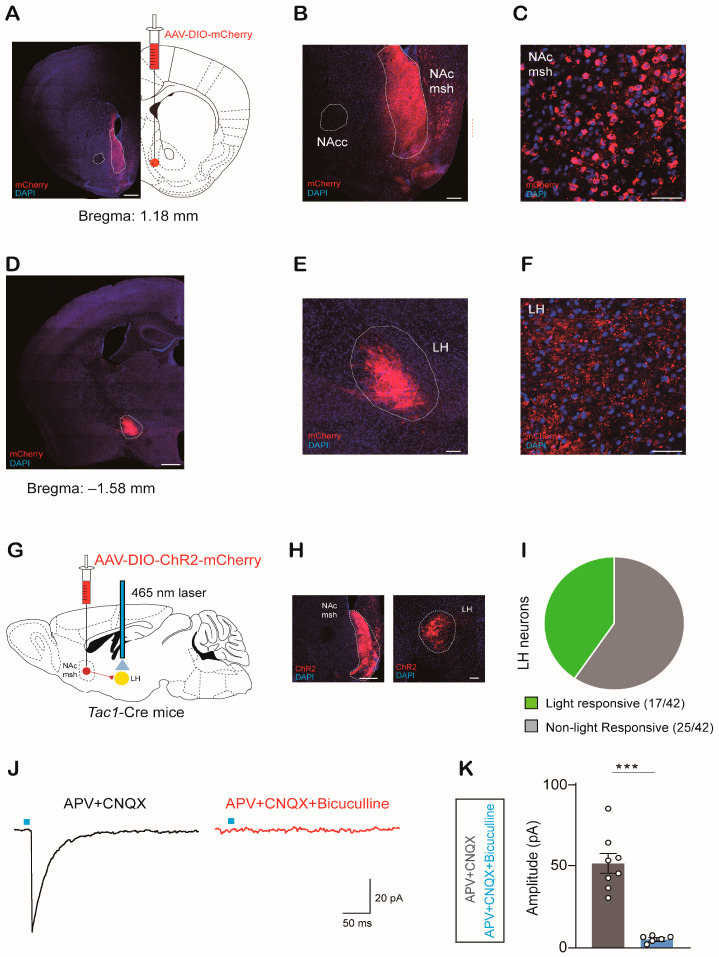
Output mapping of Tac1 neurons in the NAc medial shell: (**A**) Representative images showing virus expression in the NAc medial shell. Scale bar: 500 µm. (**B**,**C**) Coronal view documenting viral expression in NAc medial shell. Scale bar: 200 µm and 50 µm, respectively. (**D**) Images of tracing output from the NAc lateral shell to LH. Scale bar: 500 µm. (**E**,**F**) Anterograde tracing of Tac1^NAc^ neurons showing fibers in the LH. Scale bar: 200 µm and 50 µm, respectively. (**G**) Schematic of strategies used to express AAV-DIO-ChR2-mCherry in Tac1 neurons. (**H**) Representative images of injection and projection sites. Scale bar: 200 µm. (**I**) Pie chart indicating that oIPSCs were recorded in 40% of cells (17 of 42 cells), and 60% were non-responsive in 3 mice. (**J**) The GABA-A receptor antagonist bicuculline totally inhibited oIPSCs in the LH neurons induced via optical stimulation. (**K**) Average amplitude of oIPSCs recorded (two-tailed paired *t*-test, *t*_12_ = 6.396, *p* < 0.0001). APV+CNQX: N = 3, n = 8; APV+CNQX+Bicuculline: N = 3, n = 6. N: animal number; n: cell number. NAcc: nucleus accumbens core; NAc msh: nucleus accumbens medial shell; LH: lateral hypothalamic area. All data are means ± s.e.m. *** *p* < 0.001.

**Figure 3 ijms-24-04346-f003:**
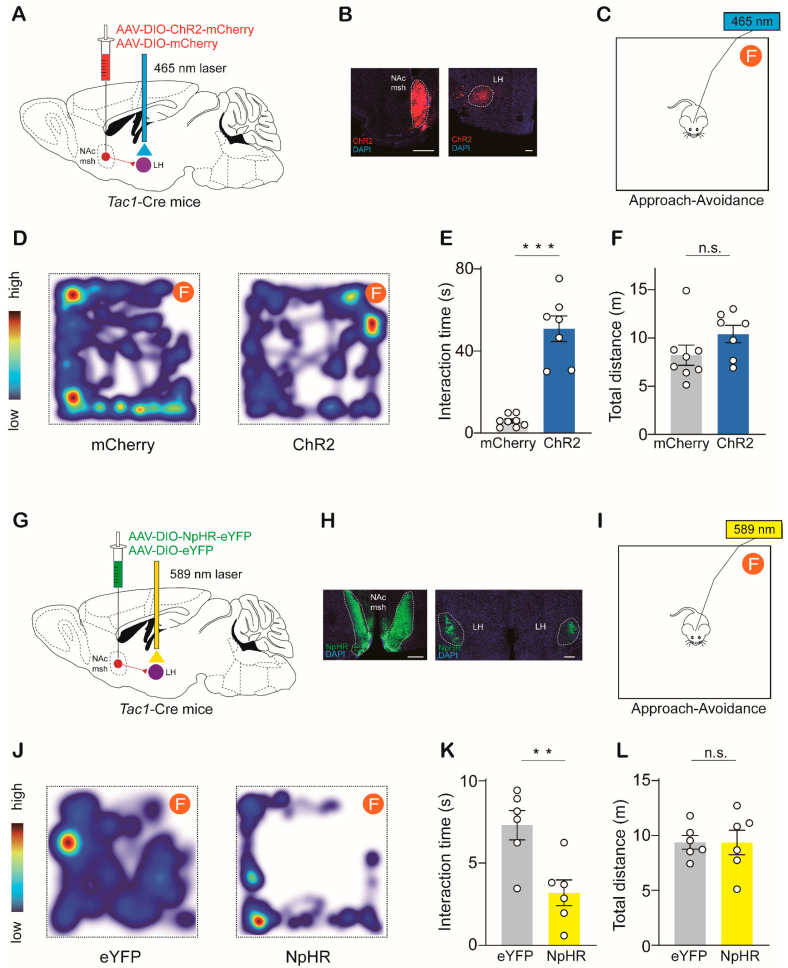
The NAc^Tac1^-to-LH pathway regulates aversive behaviors in mice: (**A**) Schematic showing unilateral optogenetic stimulation of NAc^Tac1^ inputs to LH by ChR2. (**B**) Representative images of injection and projection sites. Scale bar: 200 µm. (**C**) Schematic of approach-avoidance assay (F: form, 5% formaldehyde solution). (**D**) Heat maps display time spent in different regions of the chamber (warmer colors indicate more time). (**E**) Mice expressing ChR2, but not mCherry, spent significantly more time interacting with formaldehyde with 20 Hz optical stimulation (two-tailed paired *t*-test, *t*_13_ = 5.885, *p* < 0.0001). (**F**) Mean total distance traveled in the chamber (two-tailed paired *t*-test, *t*_13_ = 1.566, *p* = 0.1415). mCherry: N = 8; ChR2: N = 7. (**G**) Schematic showing bilateral optogenetic inhibition of NAc^Tac1^ inputs to LH by NpHR. (**H**) Representative images of injection and projection sites. Scale bar: 200 µm. (**I**) Schematic of approach-avoidance assay (F: form, 5% formaldehyde solution). (**J**) Heat maps display time spent in different regions of the chamber (warmer colors indicate more time). (**K**) Mice expressing NpHR, but not eYFP, spent significantly less time interacting with formaldehyde with continuous optical stimulation (two-tailed paired *t*-test, t10 = 3.475, *p* = 0.006). (**L**) Mean total distance traveled in the chamber (two-tailed paired *t*-test, t10 = 0.01222, *p* = 0.9905). eYFP: N = 6; NpHR: N = 6. N: animal number. NAc msh: nucleus accumbens medial shell; LH: lateral hypothalamic area. All data are means ± s.e.m. ** *p* < 0.01, *** *p* < 0.001; n.s.: not significant.

**Figure 4 ijms-24-04346-f004:**
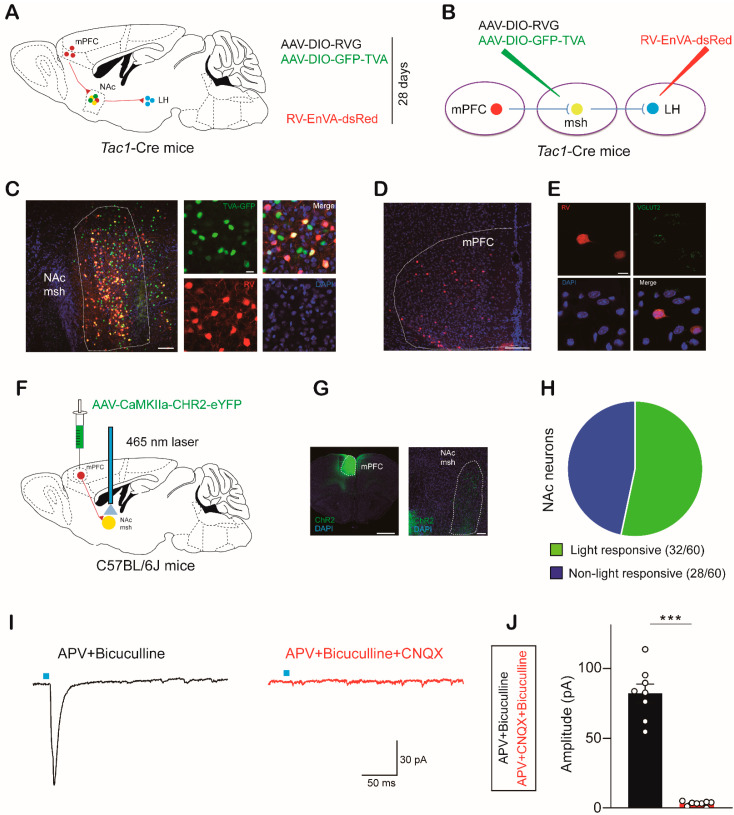
Monosynaptic viral tracing identifies the inputs of LH-projecting NAc^Tac1^ neurons: (**A**,**B**) Schematic of rabies-based cell type-specific monosynaptic tracing procedure. (**C**) Representative images showing NAc^Tac1^ starter cells (blue: DAPI; red: RV-ENVA-ΔG-dsRed; green: AAV-EF1α-DIO-EGFP-TVA. Scale bar: 100 µm and 20 µm, respectively. (**D**) Representative images showing a substantial rabies-deRed signal from LH-projecting NAc^Tac1^ neurons in the mPFC. Scale bar: 200 µm and 20 µm, respectively. (**E**) Representative mPFC image showing retrograde labeling in the mPFC from NAc msh Tac1 neurons, co-labeled with VGLUT2. (red: rabies-dsRed; green: VGLUT2; blue: DAPI. Scale bar: 10 µm). (**F**) Schematic of strategies used to express AAV-CaMKIIα-ChR2-eYFP in C57 neurons. (**G**) Representative images of injection and projection sites. Scale bar: 1 mm and 100 µm, respectively. (**H**) Pie chart indicates that oEPSCs were recorded in 53% of cells (32 of 60 cells), and 47% were non-responsive in 5 mice. (**I**) The AMPA receptor antagonist CNQX totally inhibited oEPSCs in the NAc neurons induced by optical stimulation. (**J**) Average amplitude of oEPSCs recorded (two-tailed paired *t*-test, *t*_14_ = 12.02, *p* < 0.0001). APV + Bicuculline: N = 3, n = 8; APV + CNQX + Bicuculline: N = 3, n = 8. N: animal number; n: cell number. mPFC: medial prefrontal cortex; NAc msh: nucleus accumbens medial shell; LH: lateral hypothalamic area. All data are means ± s.e.m. *** *p* < 0.001.

**Figure 5 ijms-24-04346-f005:**
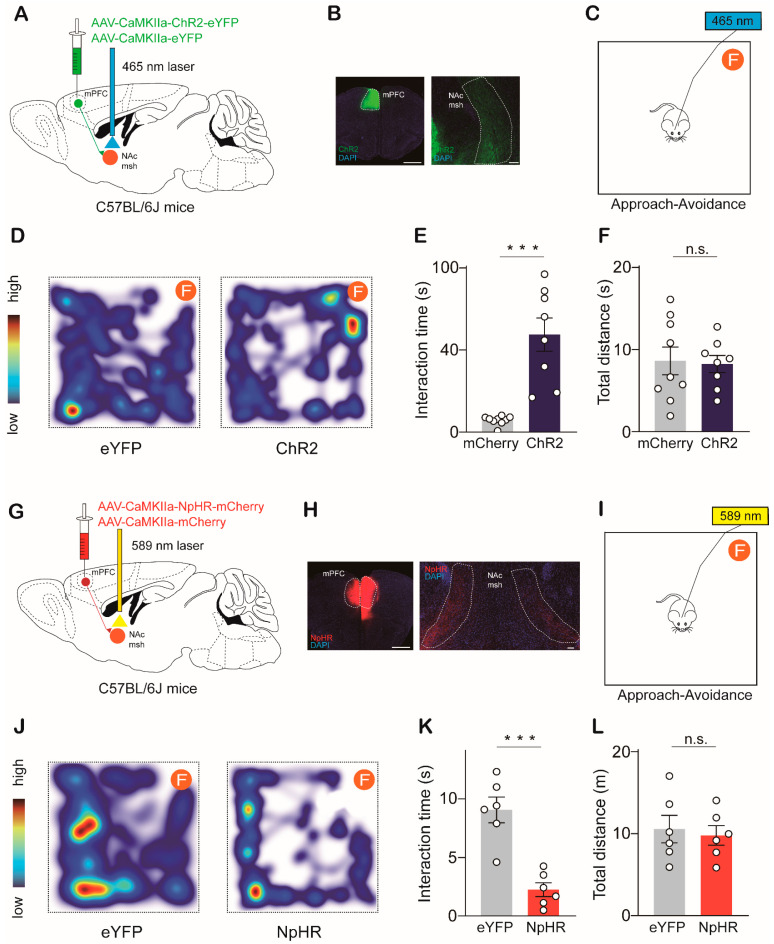
The mPFC^Glut^ inputs to NAc modulate aversive behaviors in mice (**A**) Schematic showing unilateral optogenetic stimulation of mPFC^Glut^ inputs to NAc msh by ChR2. (**B**) Representative images of injection and projection sites. Scale bar: 1 mm and 100 µm, respectively. (**C**) Schematic of approach-avoidance assay (F: form, 5% formaldehyde solution). (**D**) Heat maps displaying time spent in different regions of the chamber (warmer colors indicate more time). (**E**) Mice expressing ChR2, but not mCherry, spent significantly more time interacting with formaldehyde with 20 Hz optical stimulation (two-tailed paired *t*-test, *t*_15_ = 5.451, *p* < 0.0001). (**F**) Mean total distance traveled in the chamber (two-tailed paired *t*-test, *t*_15_ = 0.1827, *p* = 0.8575). eYFP: N = 9; ChR2: N = 8. (**G**) Schematic showing bilateral optogenetic inhibition of mPFC^Glut^ inputs to NAc msh by NpHR. (**H**) Representative images of injection and projection sites. Scale bar: 1 mm and 100 µm, respectively. (**I**) Schematic of approach-avoidance assay (F: form, 5% formaldehyde solution). (**J**) Heat maps displaying time spent in different regions of the chamber (warmer colors indicate more time). (**K**) Mice expressing NpHR, but not eYFP, spent significantly less time interacting with formaldehyde with continuous optical stimulation (two-tailed paired *t*-test, t10 = 5.493, *p* = 0.0003). (**L**) Mean total distance traveled in the chamber (two-tailed paired *t*-test, t10 = 0.3778, *p* = 0.7135). eYFP: N = 6; NpHR: N = 6. N: animal number. mPFC: medial prefrontal cortex; NAc msh: nucleus accumbens medial shell. All data are means ± s.e.m. *** *p* < 0.001; n.s.: not significant.

## Data Availability

The authors confirm that the data supporting the findings of this study are available within the article and its Appendix A. Raw data that support the findings of this study are available from the corresponding author, upon reasonable request.

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
