# Peer review of "A Nucleus Accumbens Tac1 Neural Circuit Regulates Avoidance Responses to Aversive Stimuli"

_ijms, 2023, doi:10.3390/ijms24054346_

Round 1

Reviewer 1 Report

The authors present an excellent manuscript which outlines several experiments outline the role and function of the medial prefrontal-nucleus accumbens-hippocampus circuitry in the context of aversion behaviors. The authors provide redundancy in the experiments in order to solidy their argument and to demonstrate that multiple methodologies will yield similar results. They are to be commended for their thorough work. A few additions to strengthen the paper would be to include a clearer impetus for performing this research. What have they found previously that has prompted this investigation and in the same manner, how with these findings be useful in the future and eventually for clinical care of humans. The authors can provide more information about the gaps in knowledge in the literature, as well as why studies in the past had conflicting results. There should be a mention of possible other studies that can further this work and further elucidate this phenomenon. The authors should discuss any aberrations from well establish methodologies previously described, or whether they were followed exactly as other experiments. Overall, it is a wonderful report that would benefit from more background and a vision from the authors of how this knowledge can be used to better the lives of humans in the future.

Reviewer 2 Report

1.      Authors are advised to confirm or provide the information on the expressed isoform of TAC1-mRNA in their mouse model and was it consistent across the experiments?

2.      Approximately 40% of the neurons are responding to opto-stimulus which is corresponding to the inhibitory outputs as indicated in Fig.2., however, what happens if the expression ratio of the ChR2 is reversed? Will the inhibitory output stay as observed?

3.      In figures 2 (J, K) and Fig (I, J); how does the total charge transfer looks, can authors provides a quantification of it for their traces as a supplementary information?

4.      Furthermore, in response to Opto-stimuli did authors quantify the Ca2+-level fluctuations among the neuronal populations corresponding to illustrated circuit and how it is corelating to the quantal synaptic responses at the synapses involved in their circuit and how it is fluctuating across the medial and lateral sides? Authors are advised to include the information to the supplementary data or modify the Fig.2.

5.      What age were the animals used for the experiments, were they corresponding to same age throughout the experimental spectrum?

6.      Did authors try any stimuli other than 20 Hz for 5ms? If so, what were the corresponding findings and conclusions? If not, why they choose to stick to used stimuli? Please describe and specify their reasons by including them to the manuscript.

7.      Authors are advised to confirm some dimensions of their findings by using one or two more aversive agents. Moreover, why didn’t authors considered to show the effects of their aversion agent controlling the expression of TAC1 in the neuronal populations of the corresponding circuit, wouldn’t that be more direct evidence for the involvement of TAC1-expressed substances in the control of aversive behaviors?

8.       Authors are expected to include the points indicted (in 5) to their discussion and expected to highlight the real-time physiology of the illustrated circuit.

9.      It is suggested that authors combine supplementary figures 1&2 and include to the main manuscript as figure 1. Also make changes accordingly to the rest of the manuscript.

10.   Please revisit the manuscript for other format related, grammatical or typos related issues.
